# Genetic Instability among Hitnü People Living in Colombian Crude-Oil Exploitation Areas

**DOI:** 10.3390/ijerph191811189

**Published:** 2022-09-06

**Authors:** Claudia Galeano-Páez, Dina Ricardo-Caldera, Luisa Jiménez-Vidal, Ana Peñata-Taborda, Andrés Coneo-Pretelt, Margarita Rumié-Mendoza, Alicia Humanez Álvarez, Shirley Salcedo-Arteaga, Gean Arteaga-Arroyo, Karina Pastor-Sierra, Pedro Espitia-Pérez, Paula A. Avilés-Vergara, Catalina Tovar-Acero, Sara Soto-De León, Hugo Brango, Osnamir Elias Bru-Cordero, Marvin Jiménez-Narváez, Elena E. Stashenko, Edna M. Gamboa-Delgado, Alvaro J. Idrovo, Lyda Espitia-Pérez

**Affiliations:** 1Grupo de Investigación Biomédicas y Biología Molecular, Universidad del Sinú E.B.Z., Montería 230001, Colombia; 2Grupo de Investigación Enfermedades Tropicales y Resistencia Bacteriana, Universidad del Sinú E.B.Z., Montería 230001, Colombia; 3Departamento de Matemáticas y Estadística, Universidad del Norte, Barranquilla 080001, Colombia; 4Facultad de Ciencias e Ingenierías, Universidad del Sinú E.B.Z., Montería 230001, Colombia; 5Grupo GNOXIC, Universidad del Sinú E.B.Z., Montería 230001, Colombia; 6Center for Chromatography and Mass Spectrometry (CROM-MASS), Universidad Industrial de Santander, Bucaramanga 680001, Colombia; 7Escuela de Nutrición y Dietética, Universidad Industrial de Santander, Bucaramanga 680001, Colombia; 8Public Health Department, School of Medicine, Universidad Industrial de Santander, Bucaramanga 680001, Colombia

**Keywords:** Hitnü people, PAH, oil exploitation, CBMN cyt

## Abstract

Oil exploitation, drilling, transportation, and processing in refineries produces a complex mixture of chemical compounds, including polycyclic aromatic hydrocarbons (PAHs), which may affect the health of populations living in the zone of influence of mining activities (PZOI). Thus, to better understand the effects of oil exploitation activities on cytogenetic endpoint frequency, we conducted a biomonitoring study in the Hitnü indigenous populations from eastern Colombia by using the cytokinesis micronucleus cytome assay (CBMN-cyt). PAH exposure was also measured by determine urine 1-hydroxypyrene (1-OHP) using HPLC. We also evaluated the relationship between DNA damage and 1-OHP levels in the oil exploitation area, as well as the modulating effects of community health factors, such as Chagas infection; nutritional status; and consumption of traditional hallucinogens, tobacco, and wine from traditional palms. The frequencies of the CBMN-cyt assay parameters were comparable between PZOI and Hitnü populations outside the zone of influence of mining activities (POZOI); however, a non-significant incremental trend among individuals from the PZOI for most of the DNA damage parameters was also observed. In agreement with these observations, levels of 1-OHP were also identified as a risk factor for increased MN frequency (PR = 1.20) compared to POZOI (PR = 0.7). Proximity to oil exploitation areas also constituted a risk factor for elevated frequencies of nucleoplasmic bridges (NPBs) and APOP-type cell death. Our results suggest that genetic instability and its potential effects among Hitnü individuals from PZOI and POZOI could be modulated by the combination of multiple factors, including the levels of 1-OHP in urine, malnutrition, and some traditional consumption practices.

## 1. Introduction

Several studies have shown the association between residential proximity to hydrocarbon exploitation areas and the health conditions of exposed populations. During oil exploration and production, several activities, such as drilling, transportation, and processing in refineries, produce a complex mixture of chemical compounds, including volatile hydrocarbons and polycyclic aromatic hydrocarbons (PAHs), which may affect the health of individuals living in surrounding areas [1]. In particular, when such activities are carried out in indigenous territories, they usually affect communities whose livelihoods are closely related to their surrounding environment [2]. In Latin America, well-known cases include the Mapuche in Argentina; the Apurina, Paumari, Deni, and Juma in Brazil; the U’wa in Colombia; the Shuar, Achuar, Quichua, Cofan, Siona, Waorani, and Secoya in Ecuador; the Maya in Guatemala and Mexico; the Yoro, Eseʼeje, Mascho-Piro, Amahuaca, and Urarina in Peru; and the Warao in Venezuela [3].

However, these are not the only affected indigenous groups. There are unwoven indigenous groups that live similar situations, and the scientific community and the public are generally unaware of such cases. Geographical difficulties associated with access to their territories; marked cultural differences (i.e., language and traditions of hunters and gatherers); and in Colombia, the situation of armed conflict, illegal mining, and the presence of illicit crops, have limited the study of environmental health problems and socioenvironmental conflicts in these groups. However, a recent review reported that oil extraction activities are associated with neoplasms among Amazonian indigenous people, especially cancer of the stomach, rectum, skin, soft tissues, kidneys, and cervix in adults and leukemia among children [4]. In this context, concerned by the low survival rates of the Hitnü people, considered a population at risk of physical extermination, a judge ordered the Colombian Ministry of Health to conduct an epidemiological study to evaluate the effects of oil exploitation activities on the health of the Hitnü people. The work described here is part of this study.

The Hitnü people comprise a small population of 658 people of seminomadic tradition, located along the Ele River banks in the Lipa’s subregion. Gradual changes in land use forced this population to settle, generating cultural impacts on the dynamics of community life that have led to its deterioration. As a result of the new settlement conditions and cultural practices, the Hitnü people have been affected by Chagas, a disease endemic in the region in which they are now settled [5]; animal protein consumption is also limited, and there are evident conditions of food insecurity. In addition to these changes, pressures have been exerted due to territorial conflicts with peasants, armed groups, and border zones established for hydrocarbon extraction. Thus, to better understand the effects of oil exploitation activities on the frequency of cytogenetic endpoints and cancer risk among populations living in the influence area of oil exploitation fields, we conducted a biomonitoring study using a cytokinesis micronucleus cytome assay (CBMN-cyt) to determine PAH exposure in urine samples from the Hitnü people by measuring 1-hydroxypyrene (1-OHP) using high-performance liquid chromatography.

We also evaluated the relationship between DNA damage and 1-OHP levels around the oil exploitation area, as well as the modulating effects of community health factors, such as Chagas infection; nutritional status; and consumption of traditional hallucinogens, tobacco, and alcohol made from traditional palms. The CBMN-cyt assay is widely used as a biomarker of chromosomal damage and genome instability in human populations, assessing DNA damage events, cytostasis, and cytotoxicity [6]. Furthermore, an increasing body of evidence supports the causal role of micronuclei (MN) induction in cancer development, so it is currently considered a predictive effect marker for cancer risk [7]. The results of this study will provide information to the Colombian authorities to inform decision making with respect to the preservation of the integrity of the Hitnü people and to guarantee their survival.

## 2. Materials and Methods

### 2.1. Population Characteristics and Sampling Site Location

The Hitnü people are 1 of 102 indigenous groups in Colombia and 1 of the 7 indigenous tribes in the department of Arauca. Their history is deeply rooted in the Colombian Orinoquia; they practiced different forms of nomadism based on the need to adapt to the ecological processes of the region [8]. 

The Hitnü indigenous people are currently located in urban and rural communities in Arauca, Arauquita, and Puerto Rondón, some of which are designated as “resguardos” (indigenous territories with political status and organization). Therefore, to investigate the effects of exposure to hydrocarbons, the communities were categorized as populations living in the zone of influence of the mining operation (PZOI) and populations outside the zone of influence of mining activities (POZOI) based on their proximity to the Ele River, which is suspected to be the main transport route of contaminants from the oil exploitation region. Residents of Las Vegas-Gavanes and Monogarra communities (“resguardo” San José de Lipa), as well as La Ilusión (“resguardo” La Vorágine), were considered PZOI. The presumed source of the crude oil contamination in this area is the Caricare oil fields, located approximately 30 km upstream from the indigenous settlements and where hydrocarbon exploration and exploitation activities began in 2006. 

The communities of Aspejená, located in the municipality of Puerto Rondón (“resguardo” Tierra Grata) and Caño Jesus and Bello Horizonte, situated in the urban area of Arauca, were categorized as POZOI (Figure 1). 

### 2.2. Human Biomonitoring

The field phase was carried out for 23 days (February and March 2021). Prior to access by scientific staff, a team of anthropologists and public health teachers conducted a phase of social fieldwork with the community. During this phase, participants were introduced to the potential risks and benefits associated with participation in the study. Additionally, permission to enter the communities was obtained from the indigenous authorities. Biological samples of the PZOI and POZOI were collected and transported for processing. A strict matching design was not possible between PZOI and POZOI due to the unique characteristics of the populations. Consequently, individuals from the POZOI were recruited considering gender, ethnicity, and socioeconomic status as homogenizing criteria between both populations.

Although age is one of the most commonly used factors for individual matching, this parameter could not be used in the present study due to the low tendency of the population to donate blood. This practice is considered incompatible with the cosmological value of blood and is associated with disease and death. 

Participants were instructed to answer a detailed, standard questionnaire that included data on health status, history of cancer, other chronic diseases, lifestyle, nutrition, smoking habits, and frequency of alcohol consumption. Considering the unique sociocultural characteristics of the Hitnü people, the survey also considered unique exposures, such as the consumption of yopo (a hallucinogenic substance of plant origin for ritual use), palm wine (obtained from *Scheela roystonea regia*), and tobacco.

Individuals from PZOI were recruited according to the following inclusion criteria: voluntary acceptance, good health status, and residence in the indigenous “resguardos” in proximity to the Ele River (defined as a settlement within the last five years). Inclusion criteria for individuals from POZOI included residence in areas outside the hydrocarbon exploitation zone and its zones of influence, being a member of the Hitnü ethnic group, and showing sociodemographic characteristics similar to those present in the PZOI. All data were organized and recorded in databases. No major differences in socioeconomic status or dietary habits between individuals from PZOI and POZOI were identified.

### 2.3. Blood and Urine Sample Collection

After obtaining informed consent, peripheral blood, and urine samples from 98 individuals were collected by venipuncture and in plastic containers, respectively: 58 from PZOI and 40 permanent residents of POZOI. Blood samples were collected by venipuncture by collecting 5 mL of blood in EDTA tubes for the CBMN-cyt assay and 5 mL in dry tubes without anticoagulant to assess the effects of Chagas infection. For the detection of Chagas infection, only 88 individuals were sampled. The remaining individuals refused to donate the additional 5 mL of required blood.

All samples were coded, stored at 4 °C, and transported to the laboratory 48 h after collection. Simultaneously to blood samples from PZOI and POZOI individuals, additional whole blood samples were collected from research personnel, transported, and processed under the same conditions. These samples were used as internal controls to detect potential effects caused by sample handling and transport to the laboratory. Furthermore, 10 mL urine samples were collected to determine 1-OHP levels by HPLC.

### 2.4. Determination of 1-OHP in Urine Samples by HPLC

For 1-hydroxypyrene extraction, ca. 10 mL urine samples were taken. The pH was adjusted to 5 by adding HCl. β-glucuronidase (20 µL) and ammonium acetate buffer (0.5 M, pH = 5) were added, and the resulting solution was shaken (10 s, 2500 rpm) and incubated at 37 °C for 16 h in the dark. Then, acetonitrile (5 mL) was added, and the mixture was stirred for 10 s. Then, NaCl (1 g) and MgSO4 (4 g) were added, and the solution was stirred for one minute in a bath at 4 °C, and the sample was centrifuged (8500 rpm, 5 min), filtered through 0.22 µm-filter, and injected into chromatographic equipment.

Analysis was performed in an ultra-high performance liquid chromatograph (UHPLC), 10,401 L (Thermo Scientific, Vanquish, Germering, Germany), equipped with a binary gradient pump (Thermo Scientific, 8315515), an automatic sample injector (WPS 300TRS), and a thermostated unit for the column (TCC 3000). The LC/MS interface was electrospray (HESI), and the analyzer was a high-resolution mass spectrometer with a Q-Orbitrap ion current detection system (Q-Exactive, Thermo Scientific, Bremen, Germany). Chromatographic separation was performed on a Hypersil GOLD Aq column (Thermo Scientific, Sunnyvale, CA, USA; 100 × 2.1 mm, 1.9 µm particle size) at 30 °C. The mobile phase was (A), an aqueous solution of 0.2%-formic acid; and (B), acetonitrile with 0.2%-formic acid. The initial gradient condition was 100% (A), changing linearly to 100% (B) (8 min), held for 4 min, and then returned to initial conditions for 1 min. The total running time was 13 min, with 3 min for a post run. The spectrometric system employed an electrospray interface operated in positive ion mode, with 3.5 kV capillary voltage. Mass spectra were acquired in the mass range of *m*/*z* 80–1000. The Orbitrap mass detector was calibrated with the certified reference solutions as follows: caffeine (C6035, Sigma-Aldrich, St. Louis, MO, USA), MRFA (M1170, Sigma-Aldrich, St. Louis, MO, USA); UltramarkTM 1621 Mass Spec. (AB172435, ABCR GmbH & Co., Karlsruhe, KG, Germany), and n-butylamine (471305, Sigma-Aldrich, St. Louis, MO, USA). Compounds were identified using full-scan ion acquisition mode; by the extracted ion currents (EIC) corresponding to the (M+ H) + ions of the compounds of interest by exact mass measurement, which was carried out with an accuracy of Δppm < 1; and by comparison of the retention times, mass spectra, and fragmentation pattern (experimental data) of the compounds with those of the certified reference materials. A certified 1-hydroxypyrene reference material (Dr. Ehrenstorfer, batch 752958, 99.5%, London, UK.) was used for quantification of the compound of interest.

### 2.5. Cytokinesis-Block Micronucleus Cytome (CBMN-cyt) Assay

The CBMN assay was performed according to the methodology previously described by Fenech (2007) [9]. In brief, heparinized whole blood (0.5 mL) was added to 4.5 mL of RPMI 1640 medium (Sigma R8758, St. Louis, MO, USA) supplemented with 2 mM L-glutamine (Sigma A5955, St. Louis, MO, USA), 10% fetal bovine fetal serum (Gibco 15000-044, Brazil), 100 μL/mL antibiotic-antimycotic (Sigma A5955, St. Louis, MO, USA), and 2% phytohemagglutinin (Sigma L8754, St. Louis, MO, USA). Cultures were incubated at 37 °C for 44 h under 5% CO_2_. After this incubation period, 6 μg/mL cytochalasin B (Sigma C6762, St. Louis, MO, USA) was added.

After incubation, lymphocytes were harvested by centrifugation at 1200 rpm for 8 min, centrifuged again, fixed in 25:1 (*v*/*v*) methanol/acetic acid, placed on a clean microscope slide, and stained with Diff-Quik stain (Medion diagnostics; 726443. Düdingen. CH). For each blood sample, 2000 binucleated (BN) cells (1000 from each of two slides prepared from duplicate cultures) were evaluated for the presence of DNA damage indices—micronuclei (MN), nucleoplasmic bridges (NPBs), and nuclear buds (NBUDs)—using bright-field optical microscopy at 200–1000× magnification. The number of apoptotic (APOP) and necrotic (NECRO) events was also scored in a minimum of 500 cells. The nuclear proliferation index (NDI) was calculated based on the proportion of mononucleated (MONO), BN, and multinucleated (MULT) cells in 500 cells. All slides were coded for blinded analysis according to the criteria proposed by Fenech et al. [10]. 

### 2.6. Seroprevalence of Trypanosoma cruzi Infection

For the detection of IgG antibodies against *Trypanosoma cruzi*, an enzyme-linked immunosorbent assay (ELISA) was used with two antigenic tests: a conventional ELISA test of total antigenic extracts (Test ELISA Chagas III^®^/GrupoBios, Santiago, Chile) and a recombinant antigen test (Chagas ELISA IgG + IgM^®^/Vircell Microbiologists, Granada, Spain), according to the recommendations of the National Institute of Health [11] and following the manufacturers’ recommendations. 

### 2.7. Medical Assessment for Malnutrition Signs

Because individual nutritional status is an important factor with respect to the variability of MN frequency and given the dietary limitations of the Hitnü population, three physicians examined the participants in search of suggestive clinical signs of malnutrition, following a standard protocol. A general inspection was performed, seeking mucocutaneous paleness, dry skin, absent subcutaneous fat, and pigmentation patches in the skin. The physicians then sought the presence of reddish-orange discoloration of hair, thinning of hair, swollen face, abdominal edema, koilonychia, and emaciation of arms or limbs. These signs were used to determine whether the participants had nutritional deficit, low weight, cachexia, marasmus, kwashiorkor; or acute, chronic, or global malnutrition.

### 2.8. Statistics Methods

All analyses were performed for the 98 participants. Categorical variables were described as percentages, and continuous variables with central tendency and dispersion measures, according to the observed distribution. Because the main outcome was a count event (number of micronuclei) and the study design was cross-sectional, prevalence ratios (PRs) were estimated using bivariate and multiple Poisson regressions. For categorical variables, the PR indicates the proportional increase or decrease in the frequency of CBMN-cyt parameters in the study group. For continuous variables, the PR represents the proportional increase or decrease in the frequency of CBMN-cyt assay parameters due to a one-unit increase in the variable of interest. Significant associations were defined as a *p*-value < 0.05. With respect to 1-OHP, censored values were presented according to limits of quantification (LOQ) = 2.5 and limits of detection (LOD) = 1.25. The censored values were adjusted to introduce this explanatory variable into the estimated models. The values detected and not quantified were adjusted according to (LOQ + LOD)/2, and those not detected were adjusted according to LOD/2. All statistical methods were performed using R statistical software (R Foundation for Statistical Computing, Vienna, Austria) version 4.1.2.

## 3. Results

### 3.1. Sociodemographic and Consumption Characteristics of the Study Population

Main demographic characteristics of individuals participating in the study are described in Table 1. In the POZOI, 45% of evaluated individuals were women, with an average age of 36.1 ± 15.5 years (18–57 years), whereas men had an average age of 29.3 ± 12.9 years. For PZOI, women accounted for 50% of the recruited population, with an average age of 21.1 ± 10.9 years (18–62 years). The average age of men in the PZOI was 27.9 ± 18.9 years. No significant difference in average age was detected between the PZOI and POZOI individuals. 

Results obtained for internal controls revealed no deleterious effects on sample viability caused by sample handling or transport. MN frequencies for this group were comparable to those reported in our previous studies [12].

Consumption habits of some traditional elements, such as yopo, tobacco, conventional alcohol, and alcohol made from wine palm, were also analyzed. Yopo is a psychotropic preparation made from the seeds of *Anadenanthera peregrina* and used for ritual ceremonies, wherein communication with spiritual beings is used to solve problems, obtain food, or discover cures for illnesses of community members [13]. Depending on the use, yopo is usually mixed with other natural products, such as banana, honey, and snail. However, Yopo use among the study populations was not comparable. According to our results, the PZOI presented with a high consumption frequency, whereas in the POZOI (Aspejená) group, use was reported only by one individual. The loss of this ancestral practice is also a product of the “westernization” of traditions and the new settlement conditions. 

Nevertheless, the results for the PZOI indicate that it is an important modulating factor of cytogenetic markers; analysis of this practice is a mandatory topic for further complementary studies on the living and health conditions of these communities. Alcohol intake mainly comprised palm wine; this drink was frequently consumed by 27.5% of the individuals from POZOI and 53.4% of individuals from PZOI. 

### 3.2. Frequency of the CBMN-cyt Assay Parameters

In the total population, analysis of the CBMN-cyt assay parameters revealed a non-significant incremental pattern for all DNA damage and cell death markers for PZOI individuals, except for NBUDs, which presented higher values for POZOI individuals (Table 2). The frequency of NPB was the only variable for which this difference was statistically significant, with higher levels in individuals from PZOI than POZOI. 

Because gender and age are among the main contributors to differences in CBMN-cyt assay parameters, we also assessed the influence of these confounders and exposure status as predictors of the baseline frequency of cytogenetic damage (Table 2). Among PZOI and POZOI individuals, only the frequencies of NBUDs were significantly influenced by gender, with female participants from PZOI presenting with higher frequencies than male participants (*p*-value = 0.04). Based on these results, we assumed a similar response between female and male participants for other DNA damage and cell death markers.

On the other hand, age did not influence the frequency of any of the evaluated CBMN-cyt parameters (data not shown). NDI values were comparable between the studied groups. 

### 3.3. Exposure to Hydrocarbons: Urine Levels of 1-OHP

PAHs comprise a broad group of organic pollutants that exhibit different types of metabolites resulting from their degradation. The determination of 1-OHP (a metabolite of pyrene) is one of the most widely used methodologies for the determination of PAH exposure [13]. Pyrene is non-carcinogenic but is present in most work environments with the potential release of PAHs. Thus, the product of its metabolic degradation, 1-OHP, in urine is considered in several scientific works as the optimal biological indicator of PAH exposure [14]. Thus, in order to estimate possible hydrocarbon exposure in the PZOI and POZOI groups, we assessed the levels of 1-OHP in the study participants (Table 3). 

The Mann–Whitney test showed that the concentrations of 1-OHP in urine were comparable between the two groups, with no significant differences. This result could be associated with a generalized exposure to PAHs in both populations, considering that the POZOIs are located in rural areas of the municipality of Puerto Rondón and some urban areas of Arauca Capital.

### 3.4. Malnutrition Signs

The dietary habits questionnaire allowed us to establish that the most consumed products among adults between 27 and 64 years and children between 5 and 11 years included cassava, plantain, and rice. The intake of animal protein was limited, as hunting of wildlife is infrequent among the study population, and bovine meat represents a cost they cannot assume due to their limited economic conditions. Traditionally, Hitnü people consume two meals per day: one in the morning and the other at noon. In addition, the consumption of palm wine or “vinete” is part of the traditional diet. “Vinete” intake is constant during the day, from early morning until late at night. Anthropological reports indicate that this prolonged consumption allows them to conceal hunger or “hanipa”, with energizing properties, and that if used at different levels of fermentation, vinete improves the daily motivation for work, promoting the union among the adult members of the community.

Besides the frequency and characteristics of the food consumed, we also assessed each participant’s nutritional status and the presence of malnutrition signs (Table 4).

A general inspection for clinical signs of malnutrition revealed that 56.1% of the total population presented with at least one suggestive sign. Among this group, 58.6% were individuals from PZOI, and 52.5% were individuals from the POZOI.

The most common sign of malnutrition in both populations was the presence of fine hair, which was observed in 50% of individuals, followed by discolored (reddish) hair.

Koilonychia, also known as “spoon nails,” a sign of nutritional deficit often associated with iron deficiency, was present in 6.9% of individuals from PZOI.

### 3.5. Chagas

In this study, 88 individuals were screened for *Trypanosoma cruzi* infection. The main demographic characteristics of this group are listed in Table 5. The adjusted Poisson regression showed that individuals positive for Chagas disease (antibody-positive) exhibited lower frequencies of MN (Figure 2) and apoptotic cells (Appendix A).

### 3.6. Influence of Sociodemographic and Exposure Characteristics on the Frequency of the CBMN-cyt Assay Parameters

Adjusted and unadjusted Poisson regression models were used to explore the influence of sociodemographic and exposure characteristics on the frequency of CBMN-cyt parameters (Figure 2, Figure 3 and Figure 4, Appendix A). This model was also used to evaluate the association between CBMN-cyt parameters and nutritional status, proximity to hydrocarbon exploitation areas, and urine levels of 1-OHP. These effects were expressed as the PR and the 95% confidence interval.

As previously reported, the PZOI exhibited a non-statistically significant increase in all biomarkers of DNA damage except for NBUDs. In line with these observations, the multivariate model identified the variable proximity to oil exploitation areas (PZOI/POZOI) as a risk factor for increased MN, NPB, NBUDs, and apoptotic cell frequencies: (PR: 1.45, 95%CI: 0.92–1.18), (PR: 2.13; 95%CI: 0.98–4.63), (PR: 1.18, 95%CI: 0.63–2.23), and (PR: 1.84, 95%CI: 1.37–2.47), respectively. This finding was also consistent with the modulation effect shown by 1-OHP levels. In the adjusted model, 1-OHP levels in urine appeared as a significant risk factor for increased MN frequencies only in individuals from PZOI (PR: 1.20, 95%CI: 1.05–1.37). A comparison between the communities under study revealed that the tendency of PZOI group members to present with higher MN and DNA damage frequencies was marked by values obtained for La Ilusión, Monogarra, and Las Vegas communities. In particular, La Ilusión presented significant statistical differences compared to all the other communities. In turn, these three communities showed the highest values for 1-OHP: Monogarra < Las Vegas < La Ilusión (Figure 5).

Another variable observed as a modulator of MN frequencies in the PZOI was the presence of at least one sign of malnutrition (PR: 1.28, 95%CI: 1.00–1.64). This pattern was only observed for this variable in this population. Consumption habits of coffee, palm wine, and tobacco also appeared to be important modulating factors of the MN frequencies in PZOI and POZOI. Tobacco smoking and coffee intake were associated with higher MN frequencies only in individuals from PZOI. For other variables, such as NPB, NBUDs, APOP, and NECRO, both consumption habits were found to play a protective role.

In the multiple analysis, palm wine consumption was found to modulate an increased risk of elevated MN, NPB, and NBUD frequencies in the total population and in PZOI; on the other hand, coffee drinkers had a decreased risk of increased NPBs and NBUDs. 

An unexpected finding was the protective role of Chagas disease with respect to the frequency of MN (Figure 2) and apoptotic cell (Appendix A).

Cell death biomarkers, such as apoptosis and necrosis, were modulated by tobacco and palm wine consumption, as well as malnutrition signs and 1-OHP levels. When evaluated in multiple analysis, palm wine consumption was identified as a risk factor for the total population and PZOI. Tobacco smokers had a decreased PR for both markers; surprisingly, malnutrition symptoms were associated with reduced PR in all populations (Appendix A).

## 4. Discussion

Our results suggest that genetic instability and its potential effects among Hitnü individuals are modulated by a combination of various factors: the proximity of oil exploitation zones within their territories, malnutrition conditions, and some traditional consumption practices.

Analysis of CBMN-cyt assay parameters revealed a non-statistically significant increase in all biomarkers of DNA damage and cell death for the PZOI, except for the frequency of NBUDs, which presented with higher values in POZOI. In PZOI individuals, we observed an increased risk for higher MN frequencies, with a PR of 1.45, compared to individuals from POZOI, with a of PR 0.55.

Data were consistent with previous reports in occupationally exposed workers [15] and residents of oil exploitation areas [16,17,18]; some reports described that areas closer to extraction wells showed more significant evidence of DNA damage than those farther away, suggesting a distance–damage relationship [19]. In our case, the Hitnü people living in PZOI have semi-nomadic habits, and a direct connection between the deposits’ distance and DNA damage was not measured.

Another result that could suggest a possible association between the variable of proximity to oil exploitation areas and genetic damage is the levels of 1-OHP in urine. Although urinary 1-OHP concentrations were comparable between PZOI and POZOI individuals, multiple analysis showed that 1-OHP levels constitute a risk factor for increased MN frequencies in individuals from PZOI, with an PR of 1.20. However, this parameter should be analyzed with moderation, as 1-OHP is an exposure biomarker representing 90% of PAH metabolites and a measure of 24 h cumulative exposure to PAHs [20], and the composition of PAH mixtures could be highly variable and complex depending on the area; therefore, in this case, PAH exposure may be overestimated or underestimated [13]. PAH exposure among indigenous people has been previously reported in Mexico [21,22] and Canada [22], with a focus on the determination of 1-OHP. To the best of our knowledge, this is the first assessment of DNA damage in a seminomadic indigenous PZOI potentially exposed to PAHs with the inclusion of MNBN and 1-OHP determination.

Epidemiological studies in Latin America indicate that living in proximity to oil fields is associated with increased rates of cancer and health problems [16,17] and that exposure to complex mixtures of chemical substances released during oil exploration and production could be involved [18]. This scenario establishes a severe risk for populations living near exploitation areas, with a particular aggravating factor in Colombia. Attacks against oil infrastructure by armed groups release significant quantities of crude oil into the environment, causing the contamination of water and soil resources and aggravating the contamination problem [23].

An interesting finding was observed in the settlement of La Ilusión. This site presented the highest frequency of MN and the highest levels of 1-OHP (Figure 5). This settlement was the only one in which a “jagüey” (well or ditch filled with water) or a lagoon was used for water supply. Wells and lagoons are especially susceptible to groundwater contamination with extractive wastes, including PAHs, heavy metals, and other organic contaminants. Therefore, there is a risk of water contamination in the La Ilusión settlement, which could constitute a risk for the residents. In the Monogarra and Las Vegas settlements, water is obtained directly from the Ele River or through “puntillos” (manual water pumps); the location of these communities downstream of the basin makes them susceptible to contamination by dispersion of affluents, possibly reflected in the levels of 1-OHP.

Another important modulating factor of MN frequency in the Hitnü population was the presence of signs of malnutrition. Presenting at least one sign of malnutrition was a statistically significant risk factor for high MN frequencies in PZOI, with 56.1% of the population presenting at least one suggestive sign. The role of micronutrients in maintaining genome stability has been reviewed in numerous articles [24,25,26]. Increasing evidence suggests that the availability of a varied and balanced sources of food and supplements (vitamins and minerals) significantly influences the cellular concentration of micronutrients necessary as cofactors for the enzymes involved in DNA synthesis and repair, as well as prevention of oxidative damage [27]. According to some authors, the genome damage caused by micronutrient deficiency is comparable to that caused by exposure to significant doses of chemical carcinogens, as well as ionizing and ultraviolet radiation [25].

Given the Hitnü people live almost exclusively on roots, such as cassava, rice, and bananas, we can presume that they suffer severe nutritional limitations that threaten their survival and increase their susceptibility to diseases related to genetic instability and impaired DNA repair capacity. This conclusion is fundamental for the Colombian authorities responsible for decision making, who will have to intervene with respect to the factors that cause the food deficit among the Hitnü people in order to guarantee their nutritional safety.

Another traditional diet-related practice of the Hitnü that was found to modulate the frequencies of DNA damage was palm wine (“vinete”) intake. Palm wine consumption is widespread among the PZOI and PZOI, constituting a daily-basis dietary component. Thus, we were interested in exploring whether this ethanol-based beverage could act as a positive modulator for our cytome parameters [28,29]. Our results showed that the ingestion of palm wine was a risk factor for MN, NPB, and NBUD frequencies. Furthermore, evidence found in human B lymphoblastoid cell lines supports our findings [30], which may relate to oxidative stress induction via ethanol consumption. Interestingly, a previous report demonstrated oxidative stress induction in Nigerian drinkers of palm wine due to increased lipid peroxidation and decreased selenium and glutathione peroxidase activity [31]. Another effect of palm wine ingestion may be cytotoxicity modulation, as we encountered in both univariate and multiple analysis. Ethanol may be related to proapoptotic/necrotic molecular signaling via oxidative stress mechanisms [32,33], which may explain the modulatory effects of palm wine consumption on APOP and NECRO frequencies. In our case, we hypothesize that the DNA damage effect evidenced among the Hitnü indigenous population may be related to the ingestion frequency of palm wine, which increases ethanol coexposure; however, we cannot exclude the possibility of secondary metabolite modulation of cytotoxicity, as the pericarp of *Roystonea regia* possesses important alkaloid, phenolic, and flavonoid contents, compounds known for their antioxidant capabilities [34], which may counteract the harmful effects of other contaminants. More studies are needed to confirm our results and to assess the true extent of palm wine consumption as a modulator of DNA damage response in individuals exposed to PAHs. Equally important, chemical speciation analyses are needed to fully characterize the phytochemical composition of palm wine.

The protective role of tobacco in preventing increased NPB, NBUDS, APOP, and NECRO frequencies in all participants was another interesting result. The genotoxic effects of tobacco smoke reported only for MN in PZOI have been described in previous studies related to oxidative stress induction and the regulation of the IL-8, IL-6, and TNF-α inflammatory cytokine pathways [35]. However, in concordance with the results obtained for NPB, NBUDS, APOP, and NECRO, other studies have suggested a protective effect of tobacco exposure via an individual adaptative response in exposed individuals. In vivo evidence has shown that glutathione is elevated in the lung epithelium of mice as a response to inhaled tobacco smoke [36]. In addition, DNA base excision capacity seems to increase this adaptative response from a mechanical point of view. In vitro studies have demonstrated that DNA polymerase β [37] and formamidopyrimidine DNA glycosylase [38] are essential enzymes that may increase the protective response toward the genotoxicity of tobacco smoke. Considering our results, we propose an adaptative response among the Hitnü from PZOI related to long-term exposure to tobacco smoke. As an important cultural tradition, tobacco leaves are frequently smoked in these indigenous communities.

The protective association of Chagas disease infection was another unexpected result that suggests a possible adaptive mechanism of infection that may be related to the antioxidant defense system. To the best of our knowledge, this is the first time that this protective association has been reported, so more in-depth studies are required to confirm these results and elucidate the biological mechanisms involved. This hypothesis is based on the following considerations: (1) *T. cruzi* infection generally involves increased Ca^2+^ influx [39] and exacerbated production of reactive oxygen species (ROS) in host cells [40]; and (2) the ROS released in response to *T. cruzi* infection cause DNA damage, such as 8-oxodeoxydeoxyguanine (8-oxodG) lesions and DNA strand breaks, which signal poly ADP-ribosylation of proteins (PARylation) [41]. Thus, in a state of chronic, untreated infection and a permanent state of exacerbated ROS production, it is possible that the host cell deploys mechanisms to cope with and overcome this state, including the expression of antioxidant enzymes, specific cofactors, and efficient repair pathways. These findings generate the need to study these markers of oxidative stress and DNA repair, as well as their relationship with markers of chromosomal instability in patients with chronic Chagas disease.

Limitations of the present study include the lack of availability of samples due to the perception of blood donation and its relationship to disease among the Hitnü population. Likewise, the reluctance to donate blood samples limited the analysis of some variables, such as yopo consumption, and resulted in the previously mentioned deficiencies in the pairing of individuals from PZOI and POZOI groups. On the other hand, although 1-OHP levels are considered a valid test for determining exposure to PAHs, limitations related to this marker included its high variability in the populations and its tendency to overestimate or underestimate exposure in 24 h. Finally, limited knowledge of the Spanish language among the Hitnü people limited communication, despite the use of translators. This restriction limited data collection through sociodemographic, consumption, dietary, and exposure surveys.

Future interventions with the goal of ensuring the survival of the Hitnü people should use a multifactorial approach. Further studies should include thorough studies on possible contamination with PAHs (including winter and summer monitoring), nutritional hygiene, infection, and parasitosis control, especially in the pediatric population.

## 5. Conclusions

Our results suggest that genetic instability and its potential effects among Hitnü individuals from PZOI and POZOI are modulated by a combination of various factors: the levels of 1-OHP in urine; the presence of at least one sign of malnutrition; and some ancestral practices of the Hitnü people, such as consumption of palm wine, tobacco, and yopo. Although the frequencies of the CBMN-cyt assay parameters were comparable between communities located inside and outside the zones of influence of oil exploitation complexes, we observed a non-significant incremental trend among individuals from the PZOI for most of the DNA damage parameters.

This trend was influenced by the values obtained in populations located downstream of the exploitation areas and with particular methods for water collection. Thus, in some of the explored areas, it is possible that the contamination of water sources used by the Hitnü people may be the main form of exposure to oil exploitation waste.

Our results confirm the social vulnerability of the Hitnü people and reveal the impact that oil exploitation activities have on the lives of this small indigenous community. Severe nutritional limitations threaten their survival and increase their susceptibility to diseases related to genetic instability and impaired DNA repair capacity. Additionally, the settlement of this nomadic indigenous group contributes to the occurrence of Chagas disease. This study contributes to the current knowledge on this impact and highlights the need to protect this population through direct actions of government agencies.

## Figures and Tables

**Figure 1 ijerph-19-11189-f001:**
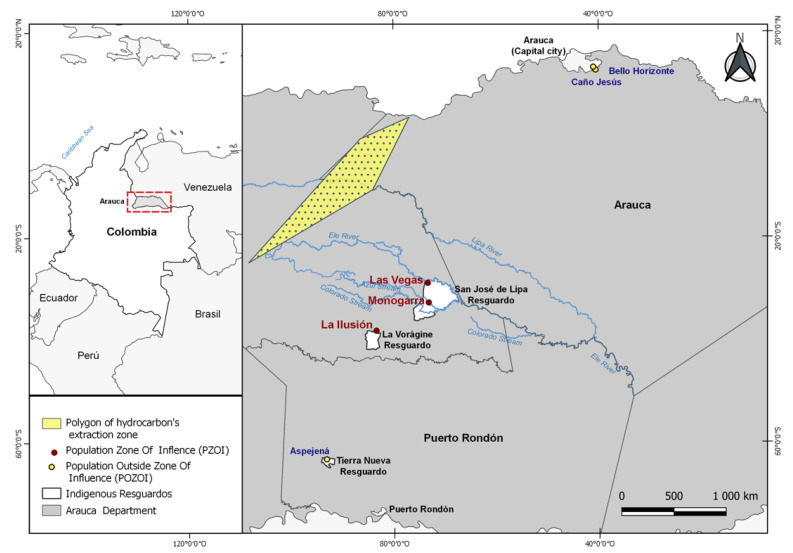
Location of Hitnü settlements in the study area.

**Figure 2 ijerph-19-11189-f002:**
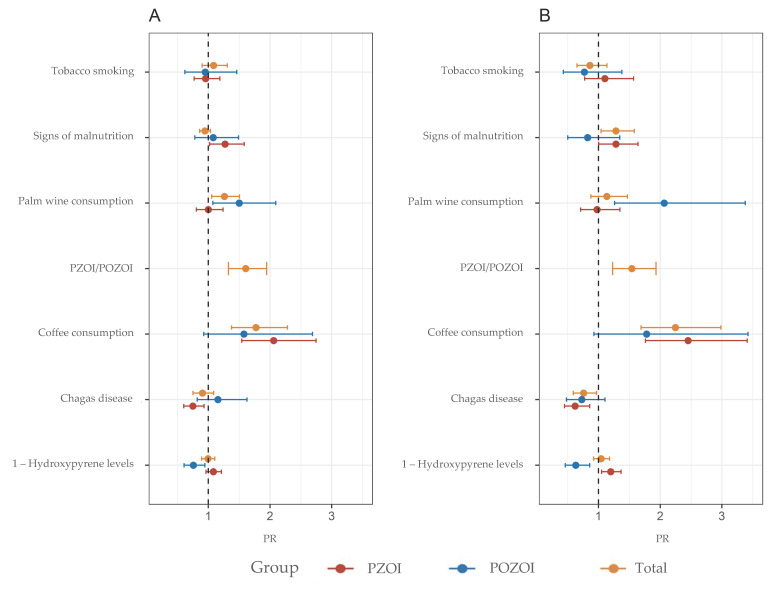
PR for MN frequency based on a Poisson regression model for PZOI and POZOI groups. (**A**) Unadjusted and (**B**) adjusted for covariates.

**Figure 3 ijerph-19-11189-f003:**
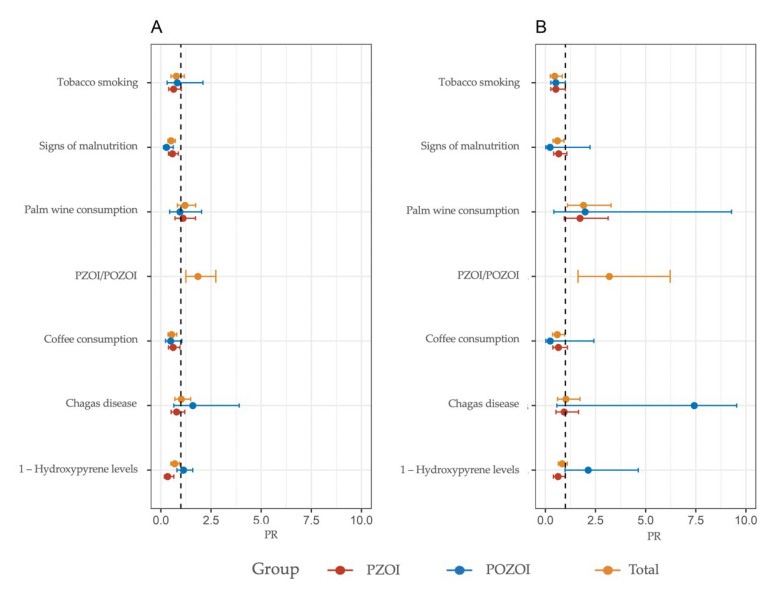
PR for NPB frequency based on a Poisson regression model for PZOI and POZOI groups. (**A**) Unadjusted and (**B**) adjusted for covariates.

**Figure 4 ijerph-19-11189-f004:**
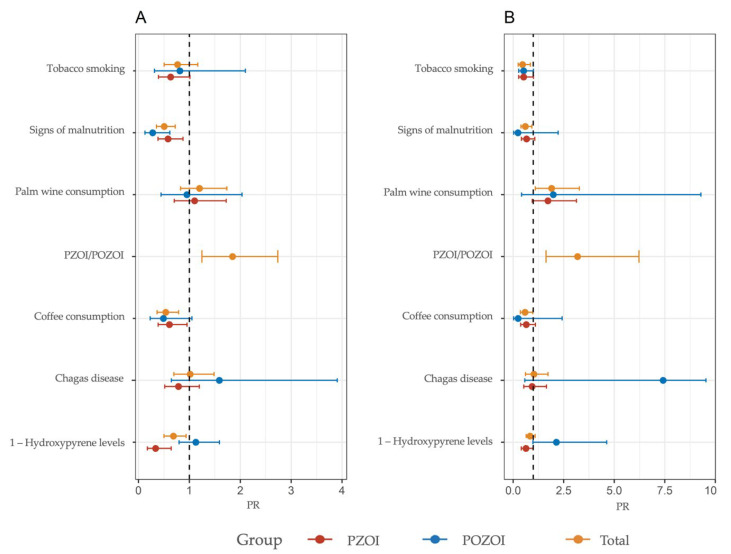
PR for NBUDs frequency based on a Poisson regression model for PZOI and POZOI groups. (**A**) Unadjusted and (**B**) adjusted for covariates.

**Figure 5 ijerph-19-11189-f005:**
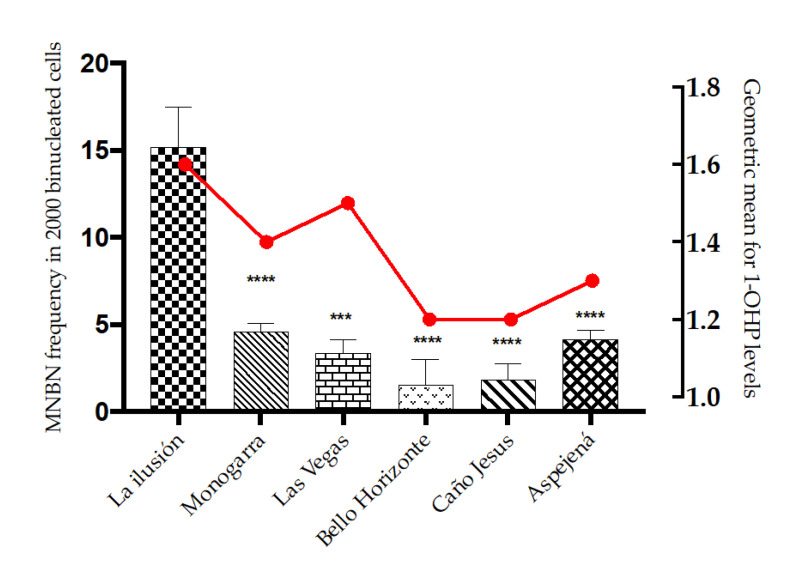
MN frequencies (bars) and 1-OHP levels (line chart) in Hitnü settlements included in the study. *** *p* ≤ 0.001; **** *p* ≤ 0.0001.

**Table 1 ijerph-19-11189-t001:** Main demographic characteristics of Hitnü indigenous people participating in the study (*n* = 98).

Group	Individuals from POZOI (Mean ± SD ^1^)	Individuals from PZOI (Mean ± SD)
Number of individuals (*n*)	40	58
Individuals by area (*n*)		
Bello Horizonte	2	
Caño Jesus	5	
Aspejená	33	
Las Vegas		29
Monogarra		18
La Ilusión		11
**Women (%)**	18 (45)	29 (50)
Age (mean ± SD)	36.1 ± 15.5	21.1 ± 10.9
**Men (%)**	22 (55)	29 (50)
Age (mean ± SD)	29.3 ± 12.9	27.9 ± 18.9
**Consumption habits (%)**		
Yopo consumers	1 (2.5)	17 (29.3)
Tobacco consumers	7 (17.5)	24 (41.3)
Palm wine (*Scheela roystonea regia*) consumers	11 (27.5)	31 (53.4)

^1^ SD: standard deviation.

**Table 2 ijerph-19-11189-t002:** Comparison of CBMN-cyt assay parameters among individuals from POZOI and PZOI.

Parameter and Variables	Individuals from POZOI			Individuals from PZOI			*p*-Value
	n	Mean ± SD	P_50_ (P_25_–P_75_)	n	Mean ± SD	P_50_ (P_25_–P_75_)	
**DNA Damage**							
**MN**							
Total population	40	3.72 ± 2.94	3.0 (2.00–5.00)	58	5.98 ± 6.35	4.0 (200–8.75)	0.22
Women	18	3.11 ± 2.65	3.0 (1.25–4.00)	29	6.24 ± 6.08	5.0 (2.00–8.00)	0.06
Men	22	4.22 ± 3.13	3.5 (2.25–5.75)	29	5.72 ± 6.70	3.0 (1.00–9.00)	0.93
**NPB**							
Total population	40	0.85 ± 1.57	0.0 (0.00–1.00)	58	1.56 ± 2.02	**1.0 (0.00–2.00)**	**0.03 ^b^**
Women	18	0.88 ± 1.49	0.0 (0.00–1.75)	29	1.79 ± 2.32	1.0 (0.00–3.00)	0.15
Men	22	0.81 ± 1.68	0.0 (0.00–1.00)	29	1.34 ± 1.69	1.0 (0.00–2.00)	0.11
**NBUD**							
Total population	40	0.30 ± 0.72	0.0 (0.00–0.00)	58	0.15 ± 0.48	0.0 (0.00–0.00)	0.30 ^b^
Women	18	0.33 ± 0.68	0.0 (0.00–0.00)	29	**0.03 ± 0.18**	**0.0 (0.00–0.00)**	**0.04 ^a^**
Men	22	0.27 ± 0.76	0.0 (0.00–0.00)	29	0.27 ± 0.64	0.0 (0.00–0.00)	0.78
**Cytostatic and Cell Death**							
**APOP**							
Total population	40	2.27 ± 3.31	1.0 (0.00–3.25)	58	5.77 ± 8.70	1.0 (0.00–9.00)	0.22
Women	18	1.72 ± 2.76	0.0 (0.00–2.75)	29	5.24 ± 8.42	1.0 (0.00–7.00)	0.21
Men	22	2.72 ± 3.70	1.5 (0.00–3.75)	29	6.31 ± 9.09	1.0 (0.00–9.00)	0.54
**NECRO**							
Total population	40	1.22 ± 2.54	1.0 (0.00–1.00)	58	1.39 ± 2.74	1.0 (0.00–1.00)	0.84
Women	18	0.61 ± 1.91	0.0 (0.00–0.00)	29	1.89 ± 3.37	0.0 (0.00–1.00)	0.12
Men	22	1.72 ± 2.91	0.0 (0.00–2.50)	29	0.89 ± 1.85	1.0 (0.00–1.00)	0.25
**NDI**							
Total population	40	2.07 ± 0.21	2.07 (1.99–2.18)	58	2.01 ± 0.26	2.0 (1.86–2.12)	0.10
Women	18	2.04 ± 0.19	2.00 (2.00–2.16)	29	2.06 ± 0.27	2.0 (1.88–2.18)	0.65
Men	22	2.09 ± 0.23	2.09 (1.97–2.22)	29	1.95 ± 0.23	1.9 (1.83–2.11)	0.07

**Bold** text indicates statistically significant values. SD: standard deviation. ^a^ Significant association in comparison to men with the same exposure status. ^b^ Significant association in comparison to individuals from POZOI.

**Table 3 ijerph-19-11189-t003:** Non-parametric Mann–Whitney test for urinary 1-OHP levels in study populations.

Group	*N*	Mean ± SD ^1^ (mg L^−1^)	G. Mean (mg L^−1^)	Min–Max (mg L^−1^)	*p*-Value
POZOI	40 (40.82%)	1.51 ± 1.00	1.31 ± 1.00	0.62–4.7	0.65
PZOI	58 (59.18%)	1.49 ± 0.94	1.35 ± 7.10	0.62–6.7	
Total	98 (100%)	1.50 ± 0.95	1.33 ± 8.98	0.62–6.7	

^1^ SD: standard deviation; G.M: geometric mean.

**Table 4 ijerph-19-11189-t004:** Non-parametric chi-squared test for malnutrition signs in the study population.

Malnutrition Signs	Individuals from POZOI	Individuals from PZOI	Total Population	*p*-Value
Signs *	21 (52.5%)	34 (58.6%)	55 (56.1%)	0.6943
Flaky and dry skin	0 (0.0%)	1 (1.7%)	1 (1.0%)	1.000
Reddish hair	3 (7.5%)	9 (15.5%)	12 (12.2%)	0.3808
Fine hair	22 (55.0%)	27 (46.6%)	49 (50.0%)	0.5375
Facial swelling	1 (2.5%)	3 (5.2%)	4 (4.1%)	0.8904
Absence of subcutaneous fat	3 (7.5%)	0 (0.0%)	3 (3.1%)	0.1281
Abdominal distension	2 (5.0%)	1 (1.7%)	3 (3.1%)	0.7424
Koilonychia	2 (5.0%)	4 (6.9%)	6 (6.1%)	1.000
Emaciation	2 (5.0%)	1 (1.7%)	3 (3.1%)	0.7424

* Individuals with at least one malnutrition sign.

**Table 5 ijerph-19-11189-t005:** Main demographic characteristics and Chagas disease seroprevalence among the Hitnü indigenous population (*n* = 88).

Group	*n* (%)	Age	ELISA Chagas	
		(Mean ± SD ^1^)	Positive *n* (%)	Negative *n* (%)
**All participants**	88 (100%)	26.38 ± 14.27	45 (51.1%)	43 (48.9%)
**Gender**				
Women	44 (50%)	22.82 ± 10.80	23 (52.3%)	21 (47.7%)
Men	44 (50%)	29.93 ± 16.41	22 (50%)	22 (50%)
**POZOI**				
Number of individuals	34 (100%)	27.35 ± 12.24	14 (41.2%)	20 (58.8%)
**Gender**				
Women	17 (50%)	25.12 ± 11.05	9 (52.9%)	8 (47.1%)
Men	17 (50%)	29.59 ± 13.28	5 (29.4%)	12 (70.6%)
**PZOI**				
Number of individuals	54 (100%)	25.76 ± 15.49	31 (57.4%)	23 (42.6%)
**Gender**				
Women	27 (50%)	21.37 ± 10.59	14 (51.9%)	13 (48.1%)
Men	27 (50%)	30.15 ± 18.35	17 (63.0%)	10 (37.0%)

^1^ SD: standard deviation.

## Data Availability

Not applicable.

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
