# Peer review of "Genetic Instability among Hitnü People Living in Colombian Crude-Oil Exploitation Areas"

_ijerph, 2022, doi:10.3390/ijerph191811189_

Round 1

Reviewer 1 Report (Previous Reviewer 3)

The authors substantially revised the manuscript, even included the new data that were not included in the original submission such as a chapter on malnutrition signs. The manuscript provides some information on factors that could influence the health of Hitnu people and have an effect on the CBMN-cytology assay parameters. To reviewer's opinion the current title of the manuscript on the part where "cancer risk" is mentioned, doesn't correspond well with the manuscript content. It is not clear why some figures have IRR label, also explanation "proxy or PR" is not clear. Proxy of what?   There is no indication of IRR in the statistical methods chapter. What does IRR represent in the study, frequency ratios in this context? Does Figures 3,4, and 5 are based on data analysis of all 98 participants? If not the authors should indicate on how many observations the multivariate Poisson regression was based on. 

Author Response

Reviewer 1:

The authors substantially revised the manuscript, even included the new data that were not included in the original submission such as a chapter on malnutrition signs. The manuscript provides some information on factors that could influence the health of Hitnü people and have an effect on the CBMN-cytology assay parameters. To reviewer's opinion the current title of the manuscript on the part where "cancer risk" is mentioned, doesn't correspond well with the manuscript content.

AUTHOR´S COMMENTS:

Thank you for your comment. We have modified the tittle to better fit the content of the manuscript, so we have removed the sentence "cancer risk". The new title is “Genetic instability among Hitnü people living in Colombian crude-oil exploitation areas”

- It is not clear why some figures have IRR label, also explanation "proxy or PR" is not clear. Proxy of what?   There is no indication of IRR in the statistical methods chapter. What does IRR represent in the study, frequency ratios in this context?

AUTHOR´S COMMENTS:

Thank you for your comment. Indeed, the multivariate Poisson regression analysis was based on 98 participants. We clarified this data in the results section.

- Does Figures 3,4, and 5 are based on data analysis of all 98 participants? If not, the authors should indicate on how many observations the multivariate Poisson regression was based on. 

AUTHOR´S COMMENTS:

Thank you for your comments. Indeed, the observations on which the multivariate Poisson regression analysis is based are 98 participants, corresponding to the individuals in the study populations.

Round 2

Reviewer 1 Report (Previous Reviewer 3)

there are no major comments. 

This manuscript is a resubmission of an earlier submission. The following is a list of the peer review reports and author responses from that submission.

Round 1

Reviewer 1 Report

This manuscript is part of a greater study regarding the assessment of cancer risks upon exposure of an indigenous population in Columbia to oil processing technologies and their consequences thereof. Although this is a fairly brief study, it is a good setting stone for the greater project it is part of. Here are some things to consider when revising the manuscript for publication in IJERPH:

Line 70: rephrase "has been endemic for several decades" (use present perfect)

Lines 74-75: ...urine samples of the Hitnu people... (reorder words)

Line 129: replace "near to" with "in the proximity of"

Line 167: Trypanosoma cruzi should NOT be italicized when it is part of an italicized subtitle

ELISA acronym must be explained (even though it is well known)

Lines 187-188: What are the adjustments made? Explain concretely!!!

Lines 195-197: Why is the average age of women subjects reported with two decimals and with standard deviation, but that of men with one decimal and no standard deviation? One decimal is more than enough given the magnitude of standard deviations. This applies also to many other numerical data reported in the manuscript with superfluous decimals.

Line 276: replace "eighty-eight" with "88" (this applies to numbers higher than 10)

Line 285: replace "supposed" with "assumed"

Line 299: replace with "proximity of deposits" 

Line 303: replace "HAPs" with "PAHs"

Line 316: "release" (plural)

Lines 319-320: rephrase "This place exhibited the highest frequency of..." 

Line 371: "it is possible"

Line 383: "supports" (singular verb)

Line 392: "corroborate" (plural verb)

Please check reference no. 25 in the list of References!!!

Results and Discussion: The main PAHs of environmental and health concern

include polynuclears above pyrene, for example benzo[a]pyrene, benz[a]anthracene, chrysene and benzo[b]fluranthene. Are these metabolized also into 1-hydroxypyrene, how does this happen and in what proportion? I find it hard to believe that this single metabolite is a sufficient biomarker for 90% of all PAHs.

Reviewer 2 Report

Here Galeano-Páez and colleagues measured the polycyclic aromatic hydrocarbons (PAH) related secondary metabolite, 1-hydroxypyrene (1-OHP) in urine samples of Hitnü people lived nearby or in oil exploitation sites by high-performance liquid chromatography. Plasma cytokinesis-block micronucleus (MN) cytome assay (CBMN-cyt) was also applied collecting biomonitoring data. They concluded that Hitnü indigenous living in the influence area of Caricare oil field in Arauca have a high risk of cancer when they are compared to unexposed populations with same socio-demographic and ethnic characteristics.

Collectively, this work is informative, and it does worth publishing within Int. J. Environ. Res. Public Health readership and community.

However, there are some minor concerns to be addressed before acceptance for publication.

To begin with, the details for HPLC measurement of urine 1-hydroxypyrene (1-OHP) concentrations are missing. Please provide adequate information and cite necessary reference.

Further the group information is misunderstanding. For the CBMN-cyt assay, there are 40 unexposed and 58 exposed subjects, totally 98, while in table 4, Chagas disease, there are 88 participants. The authors should explain the rationale of this study design.

Finally, in table 3, why the parameters and variables in 5 different groups are not the same? What are the criteria to define these parameters?

Collectively, a Major Revision is recommended for this fabulous study!

Reviewer 3 Report

The study is addressing an important issue on health of the Hitnu indigenous population in eastern Colombia in relation to potential exposure to crude-oil  as a result of oil exploitation activities in the area. The study reports on several cytogenetic end-points frequencies and prevalence ratios (PR) in exposed and unexposed groups of the Hitnu population. Although the micronuclei (MN) frequency in peripheral blood lymphocytes could be considered as a predictive factor of cancer risk, suggesting that increased MN formation is associated with early events in carcinogenesis, the issues such as interindividual variability, and possible interaction between several risk factors are not sufficiently addressed in the study. The results presented are not sufficient to conclude on cancer risks in the study population making important its further medical and epidemiological follow-up. It is an overstatement to conclude about high cancer risk in the exposed subjects based the inference mainly on the increased frequency of MN. The results are also indicating on the significant positive association between MN frequency and Yopo consumption which is not adequately discussed in the paper.  

However, the study results could be used to inform the Colombian medical authorities on the need of health monitoring and preventive countermeasures for the indigenous Hitnu population living in proximity the exposure source.

Specific comments:       

1. P.2, line 84: it is unclear what "a large body of epidemiological and ethnographic evidence" is about? evidence on what?

2. Figure 1: it is impossible to distinct between the shadows of grey on the figure, the triangles are almost invisible. 

3. please include a section in the Methods about 1-Hydroxyperen (1-OHP)measurements in urine, the result of which were used in the data analysis. Please include the results of the measurements (mean, median, max) in the Results section.

4. The authors claimed in the methods that study participants were matched by age within 5 year interval. But in Table 1, the difference in mean age of women from unexposed and exposed groups are 15 years. How it is possible if age was one of the matching factors?

5. It is unclear if Poisson regression analysis was univariate or multivariate. In other words, it is unclear if the prevalence ratios were adjusted for effects of potential confounders such as age, sex etc. Please clarify. 

6. What was the unit of increment/ decrease in the proximity to exposure area variable? 1 m, 1 km or something else? Same for the unit of increment in 1-OHP level?

7. p. 5, line 179: please clarify, how the stratification by exposure group could help "to improve the control of confounders"?  

8.  Please include in the Results, section 3.1. some description of the results in relation to Yopo consumption in the study population, as the proportion of Yopo consumers in the exposed group is almost 12 times higher than in the unexposed.

9. Please check the concordance of the results presented in the tables and corresponding text. There is a mistake in Table 2 reporting on NBUD frequencies in exposed. Same for Table 3, where the PR of 1.24 for NECRO cells that presented in the text (p.8, line 263), comes from?   

10. Table 3: why for NPB the PR estimates in relation to proximity to the exposure area were reported for total group and also for unexposed and exposed group separately, when for the remaining parameters in the table the PR estimates are reported only for the total group?   

11. P8 line 278: please refer to Table 3 for the lower frequencies in Chagas positive individuals. 

12. Table 4. The median of what is reported in the table?

13. The limitations of the study should be described in the Discussion.

14. Please check the References as several of them either are not mentioned in the text at all or a wrong reference number is cited. Some references like Ref 25 is incomplete. To reviewer's opinion it is also not appropriate to refer to the papers that were only submitted and not yet being accepted for publication.

15. The Discussion requires a substantial revision for appropriate reflection of the study findings. Somewhat higher frequencies of CBMN-cyt assay parameters in exposed group compared to the unexposed group, were not statistically significantly different between the groups, except for NPB in both genders combined. None of the parameters showed a positive association with 1-OHP levels in urine, except MN in the exposed group. Also no increased PR were observed in relation to 1-OHP levels in urine used as a continuous variable (Table 3, total participants). A proximity to the exposure area can be considered only as a surrogate measure of exposure, showing a positive association with frequency of few CBMN-cyt parameters, but lacking of evidence of association when stratified by exposure status (exposed vs unexposed).  

16. the Discussion is lacking discussion of the Yopo consumption effect on the CBMN-cyt parameters.

17. It is unexpected to see that out of all the results on association between the palm wine consumption and frequency of CBMN-cyt parameters, the authors only discussed a decreased PR for MN in the exposed group, while the PRs significantly elevated for NBUD, APOP and NECRO frequencies in relation to the palm wine consumption are lacking the discussion. The results on tobacco consumption look much more consistent across the cytological parameters and exposure groups.   

18. Conclusion, please revise/ downplay the statements on high cancer risks,  because the study results are not sufficient for a definitive conclusion on cancer risk in the study population. More research data are needed.